# Genome-wide association study identifies 16 genomic regions associated with circulating cytokines at birth

Yunpeng Wang[1,2,3,4]*, Ron Nudel[1,2], Michael E. Benros[1,5,6], Kristin Skogstrand[1,7], Simon Fishilevich[8], iPSYCH-BROAD[¶], Doron Lancet[8], Jiangming Sun[1,2], David M. Hougaard[1,7], Ole A. Andreassen[3], Preben Bo Mortensen[1,9], Alfonso Buil[1,2], Thomas F. Hansen[1,2,10], Wesley K. Thompson[1,2,11], Thomas Werge[1,2,12]*

1 The Lundbeck Foundation Initiative for Integrative Psychiatric Research, iPSYCH, Denmark, 2 Institute of Biological Psychiatry, Mental Health Center St. Hans, Mental Health Services Copenhagen, Denmark, 3 Norwegian Centre for Mental Disorders Research (NORMENT), Institute of Clinical Medicine, University of Oslo, and Oslo University Hospital, Norway, 4 Lifespan Changes in Brain and Cognition (LCBC), Department of Psychology, University of Oslo, Norway, 5 Mental Health Center Copenhagen, Copenhagen University Hospital, Denmark, 6 Department of Immunology and Microbiology, Faculty of Health and Medical Sciences, University of Copenhagen, Denmark, 7 Danish Centre for Neonatal Screening, Department of Congenital Diseases, Statens Serum Institut, Denmark, 8 Department of Molecular Genetics, Weizmann Institute of Science, Israel, 9 Department of Economics and Business Economics-National Centre for Register-based Research, University of Aarhus, Denmark, 10 Danish Headache Center, Department of Neurology, University Hospital Copenhagen, Denmark, 11 Division of Biostatistics, Department of Family Medicine and Public Health, University of California, San Diego, California, United States of America, 12 Department of Clinical Medicine, University of Copenhagen, Denmark

¶ Names of the iPSYCH-BROAD collaborators are provided in Acknowledgements.
* Yunpeng.wang@psykologi.uio.no (YW); Thomas.Werge@regionh.dk (TW)

**Data Availability Statement:** All association summary statistics will be made available upon acceptance at the iPSYCH webpage, http://ipsych. dk.

## Abstract

Circulating inflammatory markers are essential to human health and disease, and they are often dysregulated or malfunctioning in cancers as well as in cardiovascular, metabolic, immunologic and neuropsychiatric disorders. However, the genetic contribution to the physiological variation of levels of circulating inflammatory markers is largely unknown. Here we report the results of a genome-wide genetic study of blood concentration of ten cytokines, including the hitherto unexplored calcium-binding protein (S100B). The study leverages a unique sample of neonatal blood spots from 9,459 Danish subjects from the iPSYCH initiative. We estimate the SNP-heritability of marker levels as ranging from essentially zero for Erythropoietin (EPO) up to 73% for S100B. We identify and replicate 16 associated genomic regions ($p < 5 \times 10^{-9}$), of which four are novel. We show that the associated variants map to enhancer elements, suggesting a possible transcriptional effect of genomic variants on the cytokine levels. The identification of the genetic architecture underlying the basic levels of cytokines is likely to prompt studies investigating the relationship between cytokines and complex disease. Our results also suggest that the genetic architecture of cytokines is stable from neonatal to adult life.

**Funding:** This study was supported by The Lundbeck Foundation (grant numbers R102-A9118, R155-2014-1724 and R268-2016-3925 and 278-2018-1411), Denmark, the Independent Research Fund Denmark (grant number 7025-00078B), the Stanley Medical Research Institute, an Advanced Grant from the European Research Council (project number 294838) and the Stanley Center for Psychiatric Research at Broad Institute and Centre for Integrated Register-based Research at Aarhus University (MEB,KS, iPSYCH-BROAD, DMH,PBM, TFH, and TW). WKT was in part supported by NIH grant R01GM104400. DL and SF were supported by a grant from LifeMap Sciences Inc. (Calfornia, USA). This research has been conducted using the Danish National Biobank resource, supported by the Novo Nordisk Foundation (SK and DMH). YW and OAA are supported by the Research Council of Norway (Dr. Wang through a FRIPRO Young Talented Grant (#302854) and a FRIPRO Mobility grant scheme (#251134)) and the UiO:Life Science Convergence Environment (4MENT). The FRIPRO Mobility grant scheme (FRICON) is co-funded by the European Union's Seventh Framework Programme for research, technological development and demonstration under Marie Curie grant agreement No. 608695. The funders had no role in study design, data collection and analysis, decision to publish, or preparation of the manuscript.

**Competing interests:** The authors have declared that no competing interests exist.

## Author summary

Inflammation is a complex process which involves different mechanisms on both the molecular and physiological levels. It is known to play a key role in a diverse group of conditions, including cancers, metabolic and cardiovascular disease, allergies, autoimmune disease, and, in some cases, neurological and psychiatric disorders as well. Studying circulating cytokine marker levels in blood is crucial to the understanding of the disease mechanism and its relation to the inflammatory response. In this study we perform large-scale analyses (N = 9,459) to investigate the genetic underpinnings of the variation in the levels of ten different cytokines at five to seven days after birth. We show that they can be distinguished by the level to which they are genetically determined, and we find 16 genetic loci (of which 4 are novel) which are significantly associated with markers' levels in blood. We additionally map the discovered loci to locations in the genome that are involved in gene regulation, thereby providing a plausible functional mechanism. We contrast our results with previous studies using adult samples and show that the genetic control of markers levels may be stable over an individual's lifespan. These results are informative not only at the basic-research level, but also at a clinical level, as these markers are routinely used in diagnostic procedures without necessarily taking into account the individual's genetic makeup.

## Introduction

Circulating inflammatory markers are essential to human health and disease [1]. An important group of small circulating proteins are cytokines. These have important roles in cell signaling in general and in modulating immune function in particular, including inducing and reducing inflammation [2] Circulating inflammatory cytokines have been implicated in many classes of diseases, including cancers [3], cardiovascular diseases [4], metabolic diseases [5], autoimmune diseases [6] and neuropsychiatric disorders [7]. Their utility goes beyond their explanatory power in disease mechanism; since measuring their blood levels is a simple procedure, they can be useful in their diagnostic and predictive power. For example, they can be used as indicators for obesity and early cancer risk factors [8]. In addition to their involvement in disease, cytokines are also involved in both physiological function *e.g.* pain [9] and mental, cognitive, or brain function [10]. The latter point might be extremely important given emerging evidence for the links between immune function and psychiatric disorders [11–13].

Despite their relevance to disease mechanism and diagnostic power, only few studies have examined the genetic architecture of circulating inflammatory markers[14–17]. Furthermore, previous studies have mainly used adult samples; thus, it is unclear whether the genetic control of inflammatory markers varies across age groups. Here, we estimate the genetic contribution to variation in the circulating cytokine marker levels at birth for: interleukin 8 and 18 (IL8 and IL18), monocytes chemoattractant protein (MCP1 aka CCL2), thymus and reactivation regulated chemokine (TARC, also known as CCL17), erythropoietin (EPO), immunoglobulin A (IgA), C-reactive protein (CRP), brain-derived neurotrophic factor (BDNF), vascular endothelial growth factors (VEGFA) and S100 calcium-binding protein (S100B). Of note, the genetics of S100B has not been previously studied.

## Results

We use data from 12,000 neonatal blood spots as part of the Danish iPSYCH Initiative[18], in which the concentrations of ten cytokines were measured using a two-step design with a

discovery sample (N = 10,000) and a replication sample (N = 2,000). Five of the ten markers, *i. e.* BDNF, IL8, IL18, MCP1 and S100B, were measured in both samples. Both discovery and replication samples included subjects tested at birth who later in life had at least one inpatient or outpatient hospital discharge code involving one or more of six psychiatric disorders: schizophrenia, bipolar disorder, depression, autism, attention-deficit/hyperactivity disorder and anorexia (**S1 Table**)[18], as well as a random population sample.

Genome-wide genotyping of DNA extracted from neonatal blood spots was accomplished using the Infinium PsychChip v1.0 array in 23 waves (for detailed protocol see Pedersen *et al.* [18]) and used to impute ~9 million 1000 Genomes Project Phase 3 SNPs. We performed two rounds of strict quality control to remove possible technical artifacts within each wave and across waves, respectively (Materials and Methods). We inferred the ancestry of each subject using both national birth register data and genomic principal component (PC) analysis. Non-Danish subjects were subsequently removed before the genetic association analyses. In total, 8,318 and 1,141 subjects were used in the discovery and replication analyses, respectively (Materials and Methods). Marker levels were log-transformed and age-residualized using a generalized additive model with 5 degrees of freedom (hereafter normalized, Materials and Methods). As expected, we observed both positive and negative correlations of the measured marker levels; correlation coefficients range from -0.06 to 0.43, but positive correlation is observed in the majority of the cases (S33 Fig). This suggests a complex regulation mechanism for immune responses.

We first estimated the proportions of the variance of marker levels accounted for by genetic variants ($h^2_{SNP}$) using restricted maximum likelihood[19]. S100B shows the highest $h^2_{SNP}$ (0.73), while EPO has $h^2_{SNP}$~0; (Fig 1A). SNP-heritabilities for the remaining eight markers range from 0.08 (BDNF) to 0.21 (IL18 and VEGFA). For each marker, $h^2_{SNP}$ was partitioned to autosomes, revealing that SNP-heritabilities of S100B, CRP, IL18 and IgA predominantly stem from the chromosomes where their coding genes are located (Fig 1B), suggesting strong cis-regulatory mechanisms. In contrast, analyses suggest disperse and polygenic trans-regulation for IL8, MCP1 and TARC (Fig 1B).

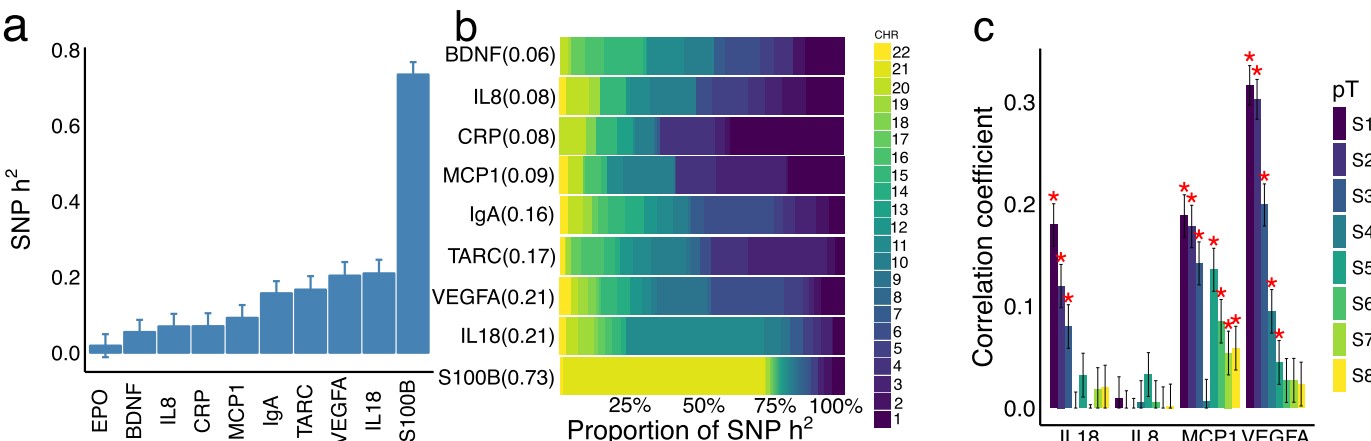

**Fig 1. SNP heritability of circulating protein levels.** The variation of circulating marker levels captured by **a.** all the genotyped SNP; **b.** SNPs on each autosome and **c.** polygenic scores computed from independent sample are shown. Cytokine SNP-heritabilities were shown in **a** by point estimates and standard errors. These point estimates were also shown in parentheses following the cytokine names on **b**. The Pearson's correlation coefficients between polygenic scores and measured protein levels in the discovery sample are stratified by different p value thresholds (pT) of association in the discovery sample (S1, P<1x10$^{-6}$; S2, P <1x10$^{-5}$; S3,1x10$^{-4}$; S4, P <0.001; S5, P < 0.01; S6, P < 0.1; S7, P <0.5; S8, P <1.0). The effect sizes used to compute polygenic scores are derived from Ahola-Olli *et al.*[17].

We correlated blood marker levels with polygenic risk scores (PGRSs) constructed from effect size estimates reported in a previous, independent study[17]. As shown in Fig 1C, moderately high correlations were observed for VEGFA, IL18, MCP1 (r = 0.31, p<$10^{-16}$; r = 0.18, p <$10^{-16}$; r = 0.19, p<$10^{-16}$; respectively, using SNPs with association p<$10^{-6}$ from the independent study), whereas IL8 blood levels and PGRSs are only marginally correlated.

We performed a genome-wide association study for each cytokine using a multiple linear regression model, including the first 6 principal components (PCs), diagnosis of any of the six disorders, genotyping wave indicators and sex as covariates (Materials and Methods). The same model was separately applied to both discovery and replication samples. We did not observe inflation in the resulting association statistics (lambda: min = 0.99, max = 1.03; S1–S10 Figs). Except for the cases of BDNF and IL8, we observed a high number genetic variants significantly associated (P<$5x10^{-9}$) with the cytokine markers, ranging from 131 for EPO to 3,941 for S100B. Extreme p values (P < $10^{-100}$) are especially common for IL18, S100B and VEGFA (Fig 2A) in line with analyses showing strong cis-regulatory mechanisms for these markers (Fig 1B). As shown in Fig 2B, common variants (minor allele frequency: MAF>20%) make up 56% of all significant variants.

We clumped the association signals into 20 independent loci (16 unique loci) indexed by at least one significant SNP (P<5 x $10^{-9}$) (Materials and Methods, S11–S30 Figs), associated with one or more markers (Table 1). Out of the 20 associations, four are novel and confirmed in the replication study (P<0.0036, Table 1). The first novel association with IL18 is in 19p13.2 indexed by rs56195122 (P = $2.4x10^{-13}$, MAF = 0.03, replication P = $6.59x10^{-4}$, S17 Fig). The SNP rs56195122 is in the first intron of the synaptonemal complex central element protein 2 gene (*SYCE2*), associated with several blood metabolites levels[21] and blood cell related traits [22, 23]. The second locus is associated with MCP1, indexed by rs4493469 (P = 1.62x10-16, MAF = 0.1, replication P = 2.0x10-3, 27kb upstream of the C-C chemokine receptor type 3 gene, *CCR3*, S20 Fig).

Genome wide significant SNPs were clumped into 20 independent regions. Information about each region includes, leading SNP (SNP), Chromosome (Chr), cytoband (Region), genomic position (Pos, hg19), effective allele (A1), alternative allele (A2), effect size (Beta), standard

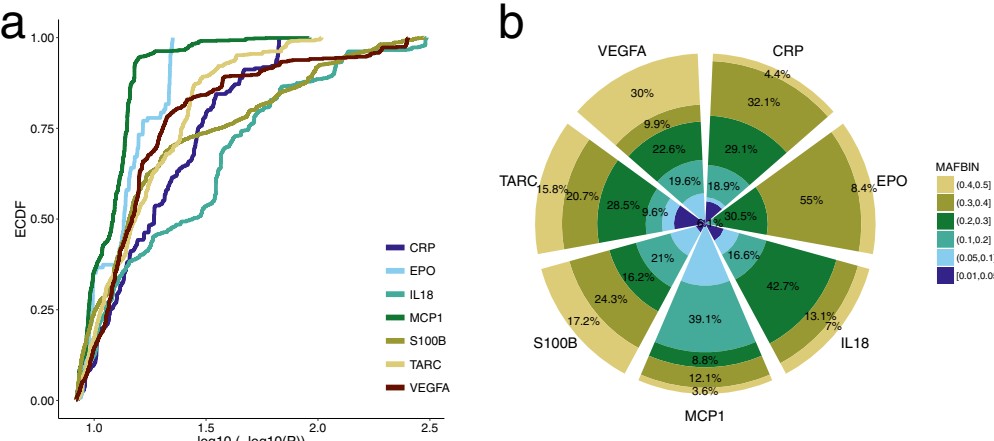

**Fig 2. Distribution of association statistics for inflammation marker level a.** The empirical cumulative distribution function of the $\log_{10}(-\log_{10}(P))$ for the association of SNPs (P<$5x10^{-9}$) with each inflammation marker. Colors indicate different markers. **b.** Distribution of SNPs (P<$1x10^{-9}$) in different minor allele frequency (MAF) bins is shown for each marker. Colors indicate different MAF intervals. Numbers in the figure shows the proportion of SNPs in the region.

**Table 1. Genome wide significant associations with blood inflammatory marker levels.**

| Marker | SNP | Chr | Region | Pos | A1/A2 | Beta | Se | P | VE | INFO | Freq | P repl | Gene |
|---|---|---|---|---|---|---|---|---|---|---|---|---|---|
| CRP | rs3091244 | 1 | 1q23.2 | 159684665 | A/G | 0.33 | 1.89E-2 | 7.47E-68 | 5.2E-2 | 1.01 | 0.32 | | CRP |
| | rs112635299 | 14 | 14q32.13 | 94838142 | T/G | -0.41 | 5.61E-2 | 3.31-E-13 | 9.02E-3 | 0.95 | 0.03 | | SERPINA1 |
| EPO | rs1130864 | 1 | 1q32.2 | 159683091 | A/G | 0.11 | 0.01 | 4.24E-23 | 1.02E-2 | 1.01 | 0.32 | | CRP |
| IgA | rs3094087 | 6 | 6p21.33 | 31061561 | T/C | 0.10 | 0.01 | 1.83E-10 | 3.12E-3 | 1.03 | 0.15 | | HLA |
| IL18 | rs10891329 | 11 | 11q23.1 | 112009892 | T/C | 0.32 | 7.4E-3 | 1E-300 | 4.0E-2 | 0.95 | 0.32 | 3.72E-37 | IL18 |
| | rs10891268 | 11 | 11q23.1 | 111301044 | A/G | 6.1E-2 | 9E-3 | 1.21E-11 | 1.26E-3 | 0.98 | 0.22 | 4.46E-4 | POU2AF1 |
| | rs56195122 | 19 | 19p13.2 | 13020506 | A/G | -0.15 | 2.08E-2 | 2.4E-13 | 1.22E-3 | 0.99 | 0.03 | 6.59E-4 | SYCE2 |
| | rs9402686 | 6 | 6q23.3 | 135427817 | A/G | 5.65E-2 | 8.3E-3 | 1.51E-11 | 1.22E-3 | 1.0 | 0.26 | 0.58 | HBS1L |
| MCP1 | rs12075 | 1 | 1q23.2 | 159175354 | A/G | 0.11 | 5.1E-3 | 1.12E-92 | 5.47E-3 | 0.98 | 0.44 | 1.84E-8 | ACKR1 |
| | rs4493469 | 3 | 3p21.31 | 46177992 | T/C | -7.01E-2 | 8.5E-3 | 1.62E-16 | 9.18E-4 | 0.97 | 0.10 | 2.0E-3 | CCR3 |
| | rs2228467 | 3 | 3p22.1 | 42906116 | T/C | -8.19E-2 | 1.01E-2 | 6.22E-16 | 8.35E-4 | 1.02 | 0.07 | 1.24E-5 | ACKR2 |
| | rs60200069 | 10 | 10q22.1 | 73503994 | T/G | -3.74E-2 | 5.2E-3 | 7.82E-13 | 6.86E-4 | 0.98 | 0.43 | 8.67E-2 | CDH23 |
| S100B | rs62224256 | 21 | 21q22.3 | 47887095 | A/G | -0.61 | 1.31E-2 | 1E-300 | 0.18 | 0.98 | 0.49 | 9.85E-38 | PCNT |
| | rs28397289 | 6 | 6p21.33 | 31197407 | T/C | 0.15 | 1.72E-2 | 7.67E-19 | 8.46E-3 | 0.98 | 0.24 | rs4713462 5.83E-5 | HLA |
| TARC | rs115952894 | 3 | 3p24.3 | 16950359 | A/G | 0.54 | 2.46E-2 | 1E-104 | 2.67E-2 | 1.0 | 0.05 | | PLCL2 |
| | rs2228467 | 3 | 3p22.1 | 42906116 | T/C | -0.41 | 2.09E-2 | 1.84E-82 | 2.05E-2 | 1.02 | 0.07 | [b]4.23E-11 | ACKR2 |
| | rs10886430 | 10 | 10q26.11 | 121010256 | A/G | 0.33 | 1.80E-2 | 1.23E-75 | 2.16E-2 | 0.89 | 0.11 | | GRK5 |
| | rs223896 | 16 | 16q21 | 57443146 | A/G | -0.12 | 1.08E-2 | 2.91E-29 | 7.10E-3 | 1.02 | 0.41 | [b]1.3E-9 | CCL17 |
| VEGFA | rs7767396 | 6 | 6q21.1 | 43927050 | A/G | 0.22 | 6.2E-3 | 3.31E-253 | 2.36E-2 | 0.99 | 0.47 | [a]3.67E-171([c]4.85E-1284) | intergenic |
| | rs11789392 | 9 | 9p24.2 | 2694914 | T/C | 0.13 | 7.0E-3 | 1.22E-73 | 8.09E-3 | 0.87 | 0.44 | [a]4.91E-5 | intergenic |

a. from Ahola-Olli et al[17]

b. from Suhre et al[16]

c. from choi et al[20].

error (Se), association p values (P), proportion of maker level variance accounted for by the leading SNP (VE), imputation quality score (INFO), association p values in the replication sample or previously reported studies (P repl) and the gene closest to the leading SNPs (Gene).

We discovered two novel regions associated with S100B levels in blood (Table 1). The first region at 21q22.3 is indexed by rs62224256 ($P<1x10^{-300}$, MAF = 0.49, replication P = 9.58x10$^{-38}$, S23 Fig) located 21kb downstream of the pericentrin gene (*PCNT*), which is a calmodulin binding protein. Remarkably, the leading SNP accounts for 18% of the variation of S100B level in blood in the discovery sample (Table 1). The A allele of the top SNP rs62224256 is associated with reduced levels corresponding to 0.32 standard deviations (SD = 0.02, $P<2x10^{-16}$) and explains 14% of S100B variation in the replication sample (Fig 3A). We also discovered that the human leukocyte antigen (HLA) region (build hg19, chr6: 28,477,797–33,448,354) is associated with the variation of circulating S100B, led by rs28397289 (P = $7.67x10^{-19}$, MAF = 0.24, S24 Fig). The association of the HLA region with S100B in the replication sample is indexed by another SNP rs4713462 (replication P = $5.83x10^{-5}$, MAF = 0.30).

Additionally, 14 of the 20 loci replicated previously-reported associations (S1 Text and S10 Table).

We constructed PGRSs for: BDNF, IL18, IL8, MCP1 and S100B, measured in both samples, for the replication analysis using the effect estimates from discovery association. Fig 3B shows Pearson's correlations between the PGRSs and the corresponding normalized marker levels stratified by different "discovery association strength". The PGRSs based on SNPs with $P<10^{-6}$ (S1) were correlated most strongly across all markers except IL8. In contrast, PGRS

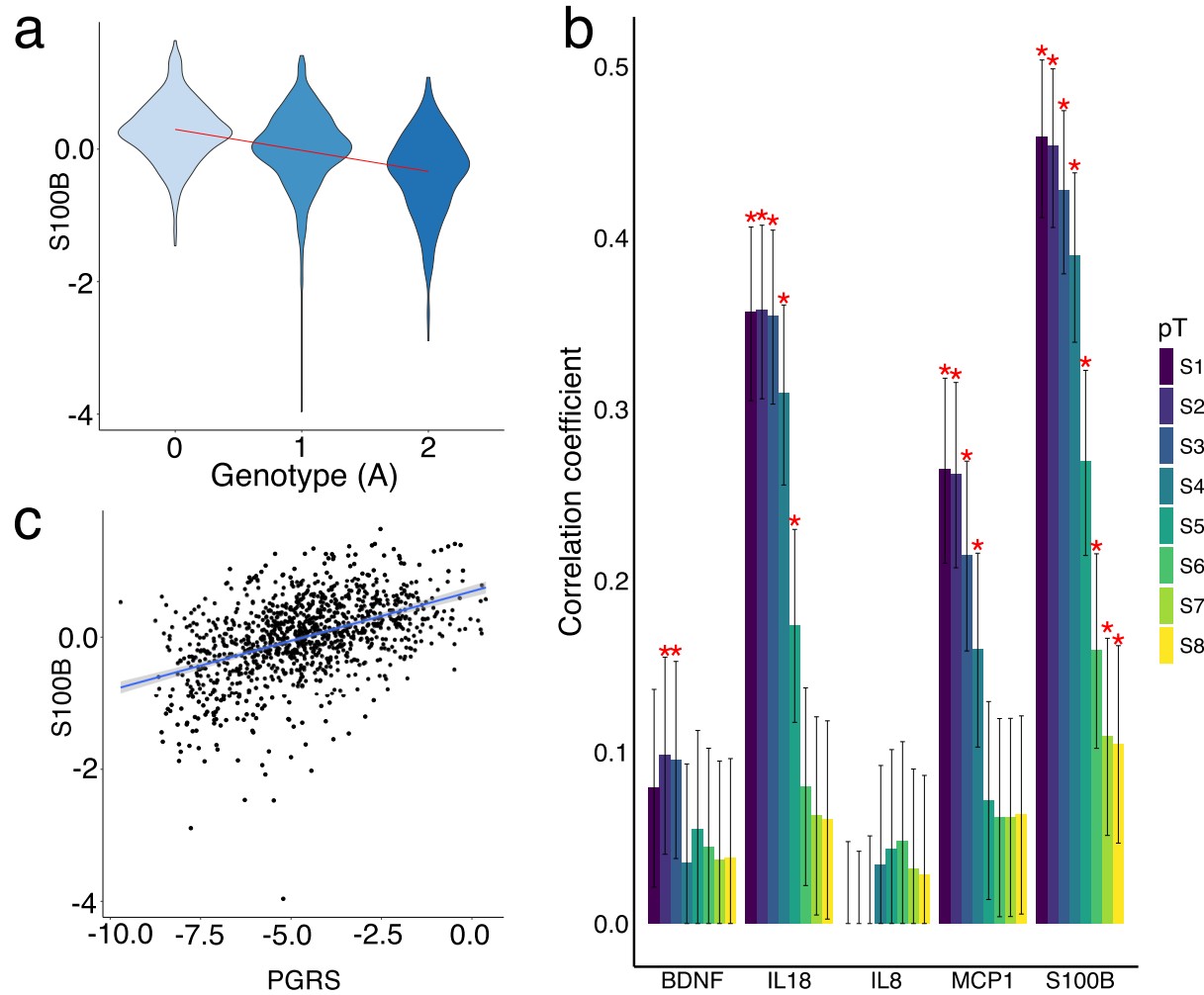

**Fig 3. Prediction of inflammation marker levels by genetic variants. a.** The distribution of the normalized S100B level in the replication sample is shown in the three genotype groups of rs62224256 (0: AA, 1, AG and 2 GG). A simple linear regression line(red) is added in the figure to show the trend. **b.** The Pearson's correlation coefficients between polygenic scores and normalized S100B level in the replication sample are stratified by different p value threshold(pT) of association in the discovery sample (S1, $P<1\times10^{-6}$; S2, $P<1\times10^{-5}$; S3, $P<1\times10^{-4}$; S4, $P<0.001$; S5, $P<0.01$; S6, $P<0.1$; S7, $P<0.5$; S8, $P<1.0$). Standard errors are show by the error bar. Stars indicate significant correlations ($P<0.00125 = 0.05/40$). **c.** A scatter plot shows the predicted S100B level (normalized, fitted strait line) in the replication sample by SNPs with $P<1\times10^{-6}$ in the discovery sample.

constructed with all SNPs (S8), *i.e.* $P\leq<1.0$, show no significant correlation except for with S100B. The PGRSs constructed with SNPs with $P<10^{-6}$ accounts for 21% of S100B variation in the replication sample (Fig 3C), and the correlation between PGRS and S100B levels is ~0.5. For comparison, an analysis based on a previously-studied discovery sample[17] is shown in S31 and S32 Figs. The observed low correlations between PGRS and IL8 levels (Figs 1C, 3B, S31 and S32 Figs) can be partially explained by the low SNP heritability for IL8 estimated in our sample (Fig 1A). On the other hand, the most significant correlations for the other cytokines was achieved by the PGRS with the lowest p-value threshold indicate that the genetic architecture of cytokines may be less polygenic than other human complex traits.

The associated loci contain a large number of genome-wide significant SNPs ($P<5\times10^{-9}$, Fig 2A), making it challenging to infer the causal variants for follow-up experimental studies. We performed Bayesian statistical fine-mapping on each associated region[24] (Materials and

Methods). For each associated region, we inferred the most probable causal configuration (causal set) assuming at most 3 causal variants per region (Materials and Methods). As shown in Table 2, eleven causal sets include their corresponding leading SNPs, among which 3 are one-variant sets. Nonetheless, 9 causal sets do not contain their corresponding leading SNPs, indicating that the top association signals may be driven by the allelic combination of SNPs in the causal sets (Table 2). Re-analysis assuming at most six causal variants per region did not change the results (S9 Table).

The FINEMAP program was applied to each associated region (500kb left and right of the leading SNP) in Table 1, assuming each region contains at most three causal variants. Abbreviations used: $\log_{10}$(BFc): common logarithm of Bayesian factor for the inferred most probable causal configuration; $\log_{10}$(BF): common logarithm of Bayesian factor for the SNP being in the causal set; PIP: posterior inclusion probability in the causal set; $R^2$: LD r-square of the SNP with the leading SNP in the corresponding region; P: association p value in the discovery sample; P repl: association p value in the replication sample or previous studies; Gene: closest gene to the corresponding SNP; Enh gene: inferred genes regulated by the corresponding enhancer.

Most of the identified genetic variants are located outside of protein-coding regions. We integrated associated loci with public epigenomic datasets[25–28] to infer plausible regulatory mechanisms. Eighteen of the 50 identified leading SNPs implicated by both association and fine-mapping analyses are located in enhancers from GeneHancer, the GeneCards Suite[29] database of human enhancers and their associated genes (Materials and Methods)(Table 2 and S2–S8 Tables). We also tested whether cytokine associated SNPs were enriched in DNAse hypertensive sites, histone modification sites and chromatin states. However, after correcting multiple testing no significant enrichment was observed (S34–S36 Figs). In Fig 4, we demonstrate the annotation by the 21q22.3 region, indexed by rs62224256, associated with S100B level. The SNPs rs11910707 (P = $1.26 \times 10^{-205}$, replication P = $1.66 \times 10^{-27}$, $\log_{10}$BF = 13.25, 12kb upstream of *PRMT2*) and rs2839314 (P = $188 \times 10^{-240}$, replication P = $4.30 \times 10^{-19}$, $\log_{10}$BF = 4.1, 22kb upstream of *DIP2A*) are the most probable causal variants. The rs11910707 SNP overlaps with the elite enhancer (Materials and Methods) GH21G046620, and rs2839314 –with GH21G046541. Both enhancers modulate the transcription of the *S100B* gene (the former through a double-elite association). Moreover, among the genes regulated by at least one of these enhancers are *PRMT2*, *DIP2A*, and *SPATC1L* (Table 2). Thus, the most highly associated signal with rs62224256 is highly likely to be a proxy of the two causal SNPs. As such, the closest gene, *PCNT*, may or may not play roles in the regulation of circulating S100B.

## Discussion

In this study, we investigate the genetic architecture of ten cytokines in whole blood at birth, in a sample of 12,000 individuals, the largest study so far. Our results highlight an important role for regulatory elements in determining levels of circulatory inflammatory markers. Importantly, we robustly replicate our findings in an in-house replication sample and by using data from other studies[16, 17]. The latter studies, in contrast to the current study, were based on adult samples, and, therefore, our results suggest that the genetic architecture of cytokines is stable from neonatal to adult life.

Inflammation and conditions associated with it, such as infections and autoimmune diseases, have been implicated in a number of disorders and medical conditions[1], including mental disorders[7, 11, 13]. In the context of the latter type of disorders, studies such as ours could be of great utility; while it has been known for a long time that mental disorders have strong genetic etiologies [30], when it comes to reliable accounts of disease mechanism, our current understanding is very limited compared to not only monogenic disorders, but also

**Table 2. Fine mapping of associated regions.**

| Marker | Leading SNP | $\log_{10}$ (BFc) | SNP | $1og_{10}$ (BF) | PIP | $R^2$ | P | P repl | Gene | Enhancer ID | *Enh Gene* |
|---|---|---|---|---|---|---|---|---|---|---|---|
| **CRP** | **rs3091244** | **93.24** | **rs3091244** | **2.56** | **0.17** | **1.0** | **7.47E-68** | | *CRP* | **GH01G159751 (rs4131568,$R^2$ = 0.82; rs12094103,$R^2$ = 0.79)** | *AIM2,CRP, FCRL6, RPL27P2,DUSP23* |
| | | | rs376195567 | 2.75 | 0.24 | 0.05 | 3.66E-24 | | *CRP* | | |
| | | | rs3093059 | 2.38 | 0.11 | 0.02 | 2.26E-14 | | *CRP* | | |
| | rs112635299 | 10.31 | rs112635299 | 3.6 | 0.63 | 1.0 | 3.31-E-13 | | *SERPINA1* | | |
| EPO | rs1130864 | 27.74 | rs1130864 | 2.59 | 0.18 | 1.0 | 4.24E-23 | | *CRP* | GH01G159751 (rs4131568,$R^2$ = 0.83; rs12094103,$R^2$ = 0.83) | *AIM2,CRP, FCRL6,RPL27P2, DUSP23* |
| | | | rs16842568 | 2.28 | 0.10 | 0.02 | 2.06E-6 | | *CRPP1* | | |
| IL18 | rs10891329 | 362.57 | rs10891325 | 3.08 | 0.55 | 0.79 | 6.19E-263 | 1.26E-23 | *SDHD* | | |
| | | | rs11214126 | 9.12 | 1.0 | 0.26 | 2.03E-98 | 4.84E-14 | *BCO2* | | |
| | | | rs10891343 | 4.74 | 0.98 | 0.55 | 7.23E-287 | 8.58E-27 | *BCO2* | | |
| | rs10891268 | 21.25 | rs10444327 | 2.65 | 0.24 | 0.0 | 1.32E-5 | 0.25 | *POU2AF1* | | |
| | | | rs117369151 | 2.53 | 0.29 | 0.05 | 1.62E-5 | 0.16 | *SIK2* | | |
| | | | rs79958943 | 3.70 | 0.78 | 0.12 | 4.49E-15 | 2.68E-3 | *SIK2* | GH19G012880 | |
| | rs56195122 | 10.44 | rs56195122 | 2.76 | 0.28 | 1.0 | 2.4E-13 | 6.59E-4 | *SYCE2* | GH19G012890 (rs2072596,$R^2$ = 0.91); GH11G111658 (rs3745647,$R^2$ = 0.90) | *GCDH,PRS6P25,ZNF709, ZNF136,ZNF788,SIK2,BGT4, C11orf88,MIR34B, MIC34C* |
| | rs9402686 | 12.10 | rs9402686 | 2.18 | 0.10 | 1.0 | 1.51E-11 | 0.58 | *HBS1L* | | |
| | | | rs56293029 | 2.09 | 0.09 | 0.93 | 2.39E-11 | 0.66 | *HBS1L* | | |
| MCP1 | rs12075 | 119.06 | rs12075 | 4.87 | 0.97 | 1.0 | 1.12E-92 | 1.84E-8 | *ACKR1* | | |
| | | | rs13962 | 0.96 | 4.64 | 0.17 | 0.013 | 0.11 | *ACKR1* | | |
| | | | rs72698561 | 3.55 | 0.65 | 0.04 | 7.85E-17 | 0.14 | *CRPP1* | | |
| | rs4493469 | 30.68 | rs6441947 | 2.91 | 0.31 | 0.03 | 0.01 | 0.61 | *CCR3* | | |
| | | | rs11923627 | 2.40 | 0.11 | 0.68 | 2.73E-15 | 2.49E-2 ([a]5.90E-4) | *CCR3* | | |
| | | | rs12495098 | 2.48 | 0.14 | 0.01 | 1.46E-10 | 1.81E-4 ([a]1.54E-20) | *CCR3* | GH03G046297 | *CCR2, CCR5,CCR1,CCRL2, TDGF1,LRRC2,FYCO1* |
| | rs2228467 | 17.78 | rs2228467 | 6.94 | 1.0 | 1.0 | 6.22E-16 | 1.23E-5 ([a]9.19E-20) | *ACKR2* | | |
| | rs60200069 | 14.15 | rs10823838 | 2.85 | 0.27 | 1.0 | 8.33E-13 | 7.90E-2 | *CDH23* | GH10G071740 | *,CCR2,PSAP, DNAJB12* |
| | | | rs3747858 | 2.76 | 0.23 | 0.03 | 5.16E-9 | 0.28 | *CDH23* | GH10G071745 | *VSIR,CDH23* |
| S100B | rs62224256 | 657.11 | rs11910707 | 13.25 | 1.0 | 0.10 | 1.26E-205 | 1.66E-27 | *PRMT2* | GH21G046620 | *PRMT2, S100B, DIP2A, SPATC1L* |
| | | | rs55912899 | 5.22 | 0.99 | 0.09 | 5.89E-129 | 1.7E-6 | *PRMT2* | | |
| | | | rs2839314 | 4.07 | 0.96 | 0.13 | 1.99E-240 | 4.30E-19 | *DIP2A* | GH21G046541 | *S100B,MCM3AP, SPATC1L, DIP2A, RNU6* |

(*Continued*)

**Table 2.** (Continued)

| Marker | Leading SNP | $\log_{10}$ (BFc) | SNP | $\log_{10}$ (BF) | PIP | $R^2$ | P | P repl | Gene | Enhancer ID | Enh Gene |
|---|---|---|---|---|---|---|---|---|---|---|---|
| **CRP** | **rs3091244** | **93.24** | **rs3091244** | **2.56** | **0.17** | **1.0** | **7.47E-68** | | *CRP* | GH01G159751 (rs4131568,$R^2$ = 0.82; rs12094103,$R^2$ = 0.79) | *AIM2,CRP, FCRL6, RPL27P2,DUSP23* |
| TARC | rs115952894 | 177.69 | rs115952894 | 4.34 | 0.92 | 1.0 | 1E-104 | | *PLCL2* | | |
| | | | rs76472873 | 3.0 | 0.36 | 0.0 | 2.21E-57 | | *PLCL2* | GH03G016916 | *MIR3713, PLCL2* |
| | | | rs369616361 | 3.78 | 0.77 | 0.14 | 0.013 | | *PLCL2* | | |
| | rs2228467 | 91.10 | rs2228467 | 7.85 | 1.0 | 1.0 | 1.84E-82 | [b]4.2E-11 | *ACKR2* | | |
| | | | rs115667394 | 2.76 | 0.31 | 0.0 | 0.02 | | *VIPR1* | | |
| | | | rs1427803 | 2.13 | 0.10 | 0.03 | 4.59E-26 | | *ACKR2* | | |
| | rs10886430 | 73.26 | rs10886430 | 13.76 | 1.0 | 1.0 | 1.23E-75 | | *GRK5* | GH10G119249 | *GC10P119246, LOC105378511* |
| | | | rs10886437 | 3.45 | 0.60 | 0.65 | 5.05E-51 | | *GRK5* | | |
| | rs223896 | 53.73 | rs4396523 | 2.94 | 0.30 | 0.11 | 2.30E-23 | | *CCL17* | GH16G057409 | *CCL17, CIAPIN1, DOK4* |
| | | | rs223897 | 2.92 | 0.29 | 0.53 | 3.51E-19 | | *CCL17* | GH16G057409 | *CCL17, CIAPIN1, DOK4* |
| | | | rs34379253 | 4.09 | 0.86 | 0.03 | 4.38E-13 | | *CCL17* | GH16G057409 | *CCL17, CIAPIN1, DOK4* |
| VEGFA | rs7767396 | 278.31 | rs9369421 | 3.70 | 0.75 | 0.0 | 0.001 | [a]1.70E-2 | intergenic | GH06G043953 | *GC06M043993, LOC105375067* |
| | | | rs73422214 | 12.65 | 1.0 | 0.07 | 1.70E-29 | [a]1.16E-13 | intergenic | GH06G043953 | *GC06M043993, LOC105375067* |
| | | | rs4481426 | 3.74 | 0.77 | 0.77 | 4.95E-194 ([c]5.25E-1060) | [a]1.24E-127 | intergenic | GH06G043953 | *GC06M043993, LOC105375067* |
| | rs11789392 | 132.60 | rs11789392 | 5.12 | 0.98 | 1.0 | 1.22E-73 | [a]4.91E-5 | intergenic | | |
| | | | rs2219143 | 4.97 | 0.97 | 0.0 | 10.8E-58 | [a]3.0E-4 | *VLDLR* | GH09G002620 | *VLDLR, PIR48978* |
| | | | rs10812148 | 3.15 | 0.38 | 0.01 | 4.33E-16 | [a]0.115 | *VLDLR-AS1* | | |

a. from Ahola-Olli et al[17]

b. from Suhre et al[16]

c. from choi et al[20].

other complex disorders such as autoimmune disorders [31, 32]. This is not necessarily due to lack of significant genetic associations, *e.g.* for schizophrenia [33], but rather it could also stem from the difficulty in defining the psychiatric traits. In this respect, leveraging the results of studies such as ours could be useful for both diagnosis and as a future avenue for research; given the links between inflammation, immunity and mental illness, and the properties of some of the inflammatory makers studied here, it could be envisaged that the latter could be used in a way similar to how endophenotypes could be used in psychiatry[34, 35]. Moreover, the intricate genetic architecture identified in this study, which highlights gene regulation, could be informative to molecular studies of psychiatric diseases and other types of diseases. For example, it is likely to prompt studies using *e.g.* Mendelian randomization[36] to investigate the relationship between inflammatory markers and complex disease.

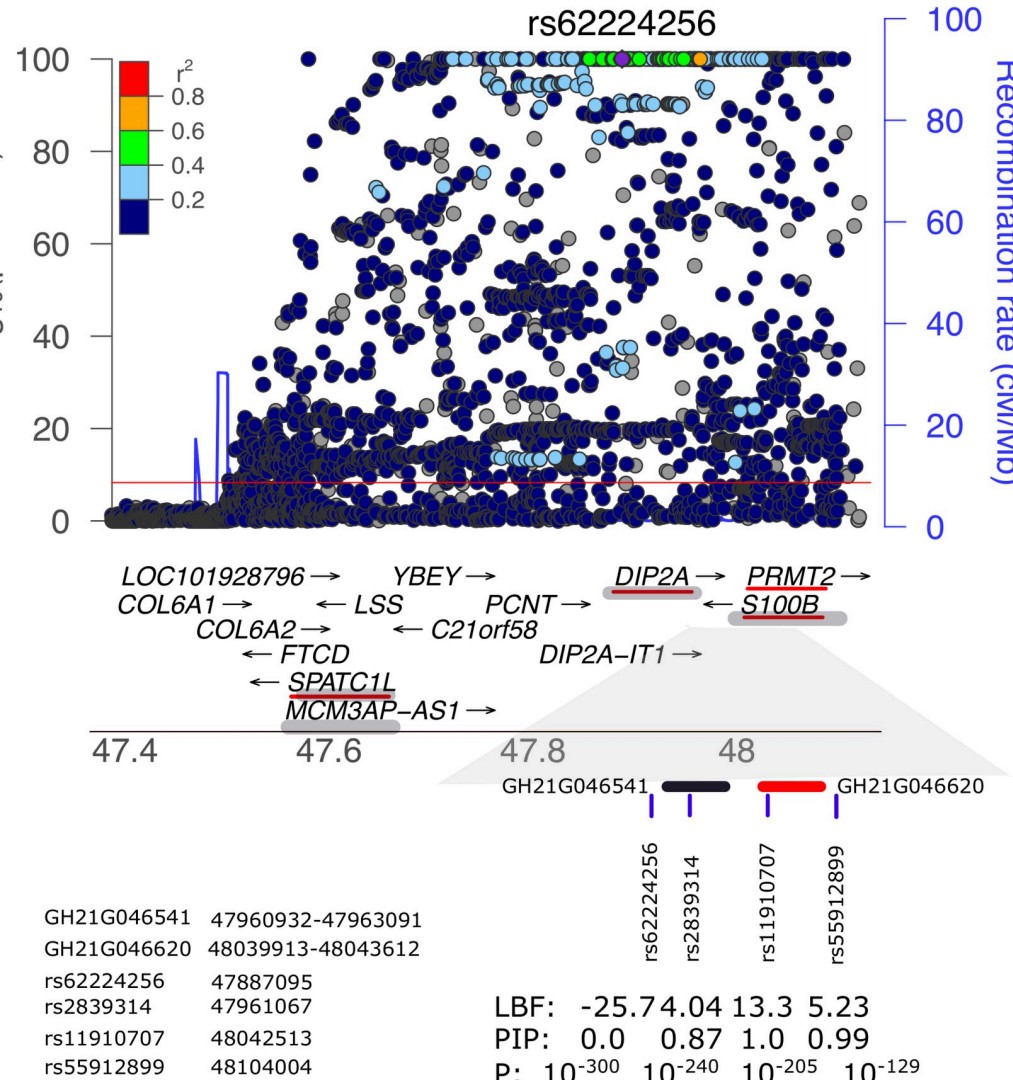

**Fig 4. Annotation of the region indexed by rs62224256 associated with S100B.** The top panel shows the regional plot. P values bellow $1 \times 10^{-100}$ were censored at $1 \times 10^{-100}$ for the clearness of illustration. Genes located in this region are shown in the middle panel. The sub-region contains rs662224256 is zoomed in approximately. Two enhancers are represented by the black and red bars. Genes regulated by the enhancers are underscored by red line and shaded bar when they are regulated by both enhancers. The $\log_{10}$ Bayesian Factor (LBF), posterior inclusion probability(PIP) of being included in the causal set and association p values (P) scales are shown in the same order as SNP rs-numbers. The genomic coordinates (build hg19) of SNPs and enhancers are shown on the lower-left panel.

The main strengths of our study are the large number of markers included, the large sample size, and the replication sample (S37 Fig). The postnatally sampling on days 5–7 day renders our findings relatively independent of the child's behavior and natural environment, which could be considered a major strength. However, it should be noted that the marker levels may, in some cases, be influenced by perinatal complications, diseases and medication administered to the child, as well as by the smoking habits, alcohol consumption, diet, weight and other general life conditions of the mother. Certain peptides, *e.g.* antibodies, cross the placenta, and neonatal levels in the child therefore reflect those of the mothers at birth, thus reducing the power of the study and accounting for the zero heritability of IL8 and BDNF. A possible source of noise in the levels of inflammatory markers is that measurements come from dried, whole

blood samples that may not precisely correspond to concentrations measures in plasma or serum in practice. However, our replication of findings from adult samples suggests that these putative biases do not present a serious limitation to the study.

In conclusion, our study sheds some light on the complex genetic architecture of inflammatory markers and highlights the important role of regulatory elements therein. We also show that the mechanisms involved are relatively stable throughout life, by comparing our results to those of studies which used adult samples. We hope that these results will prompt future studies looking into the links between inflammation and complex diseases and, in particular, that they will contribute to investigations into the mechanisms of mental illness, which have proven difficult to explain from a molecular perspective.

## Materials and methods

### Sample

The sample was based on complete and consecutive birth cohorts of all singletons born in Denmark between May 1, 1981 and December 31, 2005. Only individuals who were residents in Denmark on their first birthday and who have a known mother (N = 1,536,309) were included. From this group, 78,000 subjects were genotyped in 23 waves by the Broad Institute using the PsychChip version 1. For the discovery sample, 10,000 subjects were randomly selected from the 23 waves of the iPSYCH initiative[18]. For the replication sample 2,000 subjects were chosen from the second wave, excluding the discovery sample (for detailed description of samples see S1 Table).

### Cytokine level measurements

The 2000 samples for replication analysis were measured using Luminex technology as described by Skogstrand *et al.*[37, 38]. The second 10 000 samples used for discovery study were measured using Meso-Scale technology as described in Skogstrand et al.[39]. Briefly, dried blood spot sample were punched as 3.2mm disks into PCR-plates (Sarstedt, 72.1981.202). 130 µl extraction buffer (PBS containing 1% BSA and 0,5%Tween-20) were added to each well, and the samples were extracted in 1 hour at room temperature on a micro-well shaker set at (900rpm). The extracts were manually moved to sterile Matrix 2D tubes (Thermo Scientific, 3232) and frozen at -80˚C. One (Luminex) or two (Meso-Scale) years later, samples were thawed and analyzed using either Luminex technology in-house assays or Meso-Scale plates printed customized for the project. Analyte concentrations were calculated from the calibrator curves on each plate using 5PL (Luminex) or 4PL (Meso-Scale) logistic regression. Analytes falling below the lowest concentration within the working range were assigned to that value.

The measured levels were first inspected for potential outliers by scatter plots. Then, each marker level was logarithm transformed and age-residualized using a generalized additive model with 5 degrees of freedom, using the R function 'gam'. The resultant data was further checked for normality and outliers.

### Quality control and imputation

Quality control, and imputation were performed for each wave separately. The quality control parameters for retaining SNPs and subjects were: SNP missingness$\leq$0.05 (before sample removal); subject missingness $\leq$ 0.02; autosomal heterozygosity deviation ($|$ Fhet $| \leq$ 0.2); SNP missingness$\leq$0.02 (after sample removal); and, SNP Hardy-Weinberg equilibrium (P $>$ $10^{-6}$). Genotype imputation was performed using the pre-phasing/imputation stepwise approach

implemented in IMPUTE2[40]/ SHAPEIT2[41](chunk size of 3 Mb and with default parameters). The imputation reference set consisted of 2,186 phased haplotypes from the full 1000 Genomes Project Phase 3. Only autosome chromosomes were analyzed.

After imputation, we identified SNPs with high imputation quality (INFO $\geq$ 0.1) and minor allele frequency (MAF > 0.01). Imputed dataset across 22 waves were merged and further quality control measures were applied (min INFO $\geq$ 0.1 and MAF $\geq$ 0.01). The best-guess genotypes were called using parameters: INFO $\geq$ 0.9 and MAF > = 0.05. The set of SNPs after linkage disequilibrium pruning (r2 $\geq$ 0.02) was used for relatedness testing and population structure analysis. PLINK[42] was used for relatedness testing. One random member of a pair of subjects with pi-hat $\geq$ 0.2 were removed. Principal component analysis was performed using EIGENSOFT[43] with the same collection of autosomal SNPs. After quality control, 8,318 subjects remained for discovery and 1,141 subjects for replication sample. In total, about 9 million SNPs were used in the association study.

## SNP heritability, $h^2_{SNP}$

The merged genotypes for discovery sample were quality controlled using the same parameter as above. Before estimating the heritability, SNPs were thinned by the PLINK[38] using the command:—*indep-pairwise 100 50 0.2*. The first 6 PCs (see next section), genotyping wave indicators and sex were used as covariates in the restricted maximum likelihood-based program BOLT-REML[19]. To estimate per-chromosome SNP heritability, SNPs located in the focal chromosome was removed and the estimated $h^2_{SNP}$ was subtracted from the whole genome estimates.

## Genome-wide association

Genome-wide association study of SNPs with inflammation marker levels were performed separately for the discovery and replication sample using a multiple linear regression model implemented in PLINK[42]. Principal components were computed separately for discovery and replication, and the first 6 principal components were used as covariates, along covariates for sex and wave indicator variables. We employed the first 6 PCs following regression analyses testing each PC and each cytokine until we reached a PC which was not associated (P>0.05) with any of the 10 cytokines. Manhattan plots in S1–S8 Figs presented the association results. The genomic inflation factors were estimated and shown in the quantile-quantile plots in S1–S10 Figs. The regional association results were constructed using LocusZoom[44](S11–S30 Figs). The phenotypic variance explained by a SNP was estimated by the $\left(\beta * \sqrt{2 * p * (1 - p)}\right)^2$, where $\beta$ is the estimated effect and $p$ the allele frequency in the discovery sample.

## Associated regions and genes

Association results were 'clumped' using PLINK based on the linkage disequilibrium structure of the 1000 Genomes projects phase 3 EUR dataset, with parameters–*clump-p1 5e-9 –clump-2 1e-6 –clump-r2 0.1*. Five hundred kilo-base (kb) were used as inter-region distance threshold. Genes whose genomic coordinates located within the boundaries of each region were assigned to the corresponding region. SNPs with the smallest association p values were taken as the leading SNP for the corresponding region. The associated SNPs were annotated to the closest genes by genomic position the Ensembl tool VEP[45] (S2–S8 Tables).

## Fine mapping

Association regions were fine-mapped using the FINEMAP[24] program. Regions were defined as genomic segments 500kb on both sides of the most significant SNP in an associated

region ($P < 5 \times 10^{-9}$). Linkage disequilibrium data from the 1000 Genomes Project phase 3 European sample were used in fine mapping. We performed two analyses: the first set the maximum number of causal variants to 3 and the other to 6. S2–S8 Tables listed all SNPs with posterior inclusion probability (PIP) $> 0.1$ for 3-causal variants analysis. S9 Table listed all inferred causal SNPs in each region for 6-causal variants analysis. The $\log_{10}$ Bayesian factors for the causal set (log10BFc) and for each SNP are shown in the tables along with association statistics.

## Enhancer annotation

The associated SNPs were mapped onto genomic enhancer regions from the GeneHancer database (v4.5) [25] using a specially-prepared annotated dataset. The GeneHancer database contains enhancers that were integrated from five enhancer sources (Ensembl[46], ENCODE [47], VISTA[48], dbSUPER[49] and FANTOM[50]) and enhancer-gene connections that are based on five methods (eQTLs[51], eRNAs[50], TF-gene expression correlations, capture-HiC [52], and genomic distance from TSS). Double-elite associations are considered to be more confident annotations and are defined as enhancer-gene connections for which both the enhancer itself and the connection to the gene are supported by at least two sources or methods, respectively.

## Polygenic risk scoring

We computed the polygenic risk scores (PGRS) for both discovery and replication samples. To compute the effect size: for discovery sample, we used the association results from the previous study[17]; and, for the replication sample, we used both the association results from discovery sample and the same previous study. The association summary statistics were first carefully filtered by removing SNPs with: MAF $< 0.05$ or INFO $< 0.8$ or having a multi-character allele. We, then, clumped the resultant data based on the 1000 Genomes Project 3 EUR linkage disequilibrium structure using the program PLINK[42] with parameters:—clump-p1 1.0,—clump-p2 1.0,—clump-r2 0.1 and—clump-kb 500. The same program was used for scoring each subject in our sample. The correlations between normalized marker levels and PGRS were computed using the R program with the *cor.test* for the Pearson' correlation. The proportions of the variance explained for each marker by each PGRS was computed as the square of the Pearson's correlation coefficients.

## Supporting information

**S1 Text. Additional analyses performed.**
(PDF)

**S1 Table. Description of samples.**
(PDF)

**S2 Table. Full annotation results for CRP level using FINEMAP, Ensemble VEP, HaploReg, Enhancer and Sherlock.**
(XLSX)

**S3 Table. Full annotation results for EPO level using FINEMAP, Ensemble VEP, HaploReg, Enhancer and Sherlock.**
(XLSX)

**S4 Table. Full annotation results for IL18 level using FINEMAP, Ensemble VEP, HaploReg, Enhancer and Sherlock.**
(XLSX)

**S5 Table. Full annotation results for MCP1 level using FINEMAP, Ensemble VEP, HaploReg, Enhancer and Sherlock.**
(XLSX)

**S6 Table. Full annotation results for S100B level using FINEMAP, Ensemble VEP, HaploReg, Enhancer and Sherlock.**
(XLSX)

**S7 Table. Full annotation results for TARC level using FINEMAP, Ensemble VEP, HaploReg, Enhancer and Sherlock.**
(XLSX)

**S8 Table. Full annotation results for VEGFA level using FINEMAP, Ensemble VEP, HaploReg, Enhancer and Sherlock.**
(XLSX)

**S9 Table. FINEMAP results for regions associated with S100B assuming six causal variants.**
(XLSX)

**S10 Table. Replication of SNPs identified by present study with those reported by Ahola-Olli et al.**
(XLSX)

**S1 Fig. The Manhattan & qq plots for BDNF level.**
(PDF)

**S2 Fig. The Manhattan & qq plots for IL8 level.**
(PDF)

**S3 Fig. The Manhattan & qq plots for CRP level.**
(PDF)

**S4 Fig. The Manhattan & qq plots for EPO level.**
(PDF)

**S5 Fig. The Manhattan & qq plots for IgA level.**
(PDF)

**S6 Fig. The Manhattan & qq plots for IL18 level.**
(PDF)

**S7 Fig. The Manhattan & qq plots for MCP1 level.**
(PDF)

**S8 Fig. The Manhattan & qq plots for S100B level.**
(PDF)

**S9 Fig. The Manhattan & qq plots for TARC level.**
(PDF)

**S10 Fig. The Manhattan & qq plots for VEGFA level.**
(PDF)

**S11 Fig. Region plot for CRP-rs3091244 association.**
(PDF)

**S12 Fig. Region plot CRP rs112635299 association.**
(PDF)

**S13 Fig. Region plot EPO rs1130864 association.**
(PDF)

**S14 Fig. Region plot IgA rs3094087 association.**
(PDF)

**S15 Fig. Region plot IL18 rs10891329 association.**
(PDF)

**S16 Fig. Region plot IL18 rs10891268 association.**
(PDF)

**S17 Fig. Region plot IL18 rs56195122 association.**
(PDF)

**S18 Fig. Region plot IL18 rs9402686 association.**
(PDF)

**S19 Fig. Region plot MCP1 rs12075 association.**
(PDF)

**S20 Fig. Region plot MCP1 rs4493469 association.**
(PDF)

**S21 Fig. Region plot MCP1 rs2228467 association.**
(PDF)

**S22 Fig. Region plot MCP1 rs60200069 association.**
(PDF)

**S23 Fig. Region plot S100B rs62224256 association.**
(PDF)

**S24 Fig. Region plot S100B rs28397289 association.**
(PDF)

**S25 Fig. Region plot TARC rs115952894 association.**
(PDF)

**S26 Fig. Region plot TARC rs2228467 association.**
(PDF)

**S27 Fig. Region plot TARC rs10886430 association.**
(PDF)

**S28 Fig. Region plot TARC rs223896 association.**
(PDF)

**S29 Fig. Region plot VEGFA rs7767396 association.**
(PDF)

**S30 Fig. Region plot VEGFA rs11789392 association.**
(PDF)

**S31 Fig. Polygenic score for discovery sample based on Ahola-Olli et al.**
(PDF)

**S32 Fig. Polygenic score for replication sample based on Ahola-Olli et al.**
(PDF)

**S33 Fig. Pearson's correlation among inflammation markers.**
(PDF)

**S34 Fig. Enrichment of genetic variants with DNAse hypersensitive sites.**
(PDF)

**S35 Fig. Enrichment of genetic variants with histone modification.**
(PDF)

**S36 Fig. Enrichment of genetic variants with chromatin state.**
(PDF)

**S37 Fig. Power analysis of discovery and replication samples.**
(PDF)

## Acknowledgments

iPSYCH-BROAD Collaborators:
Andrew J. Schork
Vivek Appadurai
Carsten Bøcker Pedersen
Marianne Giørtz Pedersen
Jonas Bybjerg-Grauholm
Marie Bækved-Hansen
Benjamin M. Neale
Mark J. Daly

## Author Contributions

**Conceptualization:** Yunpeng Wang, Michael E. Benros, Thomas F. Hansen, Wesley K. Thompson, Thomas Werge.

**Data curation:** Yunpeng Wang, Kristin Skogstrand, Jiangming Sun, Alfonso Buil, Thomas F. Hansen.

**Formal analysis:** Yunpeng Wang, Jiangming Sun, Wesley K. Thompson.

**Funding acquisition:** Yunpeng Wang, David M. Hougaard, Ole A. Andreassen, Preben Bo Mortensen, Wesley K. Thompson, Thomas Werge.

**Investigation:** Yunpeng Wang, Ron Nudel, Kristin Skogstrand, Jiangming Sun, David M. Hougaard, Alfonso Buil.

**Methodology:** Yunpeng Wang, Kristin Skogstrand, Jiangming Sun, Alfonso Buil, Thomas F. Hansen, Wesley K. Thompson.

**Project administration:** Ole A. Andreassen, Preben Bo Mortensen, Thomas Werge.

**Resources:** Ron Nudel, Simon Fishilevich, Doron Lancet, David M. Hougaard, Preben Bo Mortensen, Thomas Werge.

**Software:** Ron Nudel, Simon Fishilevich, Doron Lancet.

**Supervision:** Michael E. Benros, Thomas Werge.

**Visualization:** Yunpeng Wang.

**Writing – original draft:** Yunpeng Wang, Ron Nudel, Michael E. Benros, Kristin Skogstrand, Simon Fishilevich, Doron Lancet, Ole A. Andreassen, Alfonso Buil, Thomas F. Hansen, Wesley K. Thompson, Thomas Werge.

**Writing – review & editing:** Yunpeng Wang, Ron Nudel, Michael E. Benros, Kristin Skogstrand, Simon Fishilevich, Doron Lancet, Jiangming Sun, David M. Hougaard, Ole A. Andreassen, Preben Bo Mortensen, Alfonso Buil, Thomas F. Hansen, Wesley K. Thompson, Thomas Werge.

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
