## [Decision Letter · Decision Letter 0]

1 Jul 2020

Dear Dr wang,

Thank you very much for submitting your Research Article entitled 'Genome-Wide Association Studies Identify 16 Genomic Regions Associated with Circulating Inflammatory Markers at Birth' to PLOS Genetics. Your manuscript was fully evaluated at the editorial level and by independent peer reviewers. The reviewers appreciated the attention to an important topic but identified some aspects of the manuscript that should be improved.

We therefore ask you to modify the manuscript according to the review recommendations before we can consider your manuscript for acceptance. Your revisions should address the specific points made by each reviewer.

[LINK]

Yours sincerely,

Caroline Relton, PhD

Associate Editor

PLOS Genetics

Hua Tang

Section Editor: Natural Variation

PLOS Genetics

This is an original and interesting manuscript. The reviewers have highlighted the need for additional clarification and technical details. It would also be of interest to elaborate on the potential overlap in genetic architecture between adult and neonatal genetic variation associated with cytokines, perhaps through the application of LD score regression.

Reviewer's Responses to Questions

**Comments to the Authors:**

Reviewer #1: This study by Wang et al. performs a genome wide association studies examining ten cytokines extracted from neonatal blood birth spots. The study is well performed and accompanied by SNP heritability, fine-mapping and through polygenic risk score analysis compares the genetic architecture in adults to levels of cytokines in newborns.

Major comments:

A power calculation has not been presented. This could be performed based on the previous cytokine GWAS by Ahola-Olli et al and indicated whether the numbers are sufficient. This could then indicate the numbers needed for the replication sample. Additionally, is cytokine data only available 12,000 individuals within the Danish iPSYCH?

The split between the discovery and replication samples needs to be explained. This appears not to be random due to the loss of almost half the individuals in the replication sample after removing the non-Danish individuals compared to only a fifth in the discovery. Equally, only half of the cytokines are present in the replication sample.

The abstract and discussion is then misleading in terms of number of individuals and number of cytokines included in this study.

There are analysis and figures in the supplementary material that are not referred to in the text at all (S33-38). These analyses would be very interesting to discuss in the results.

The introduction and discussion should be expanded. They could include for example:

Introduction: why choice of those particular markers; what the markers are; merits of performing GWAS; etc. Discussion: not a proper replication; clinical relevance of knowing an individual’s genetic predisposition to cytokines; PGRSs only correlating for certain cytokines; etc.

In the inflammation marker level measurements, the samples with concentrations falling below the lowest concentration within the working range should be excluded rather than being assigned to that value.

The polygenic risk scores need more of an explanation: it is unclear why only specific ones were generated. I’m not convinced presenting it in terms of pvalue thresholds adds to the conclusions.

There needs to be more explanation about the conclusion of cis regulatory mechanisms from the proportion of SNP heritability from each chromosome and the extreme p values.

Please double check the colour scheme is suitable for colour blind people.

Results:

The individual figures from the supplementary need to be referenced in the results rather than just ‘Appendix S1’.

Line 110 and 111: should be SNP heritabilities.

Line 113: How much of the SNP heritability stems from the coding genes? Because if that explains the SNP heritability then you can’t conclude that there are strong cis-regulatory mechanisms.

Line 114: ‘analyses suggest’ is surplus.

Line 140: ‘clumped’ needs a better explanation

Methods:

The discovery and replication p value needs to be justified.

Is there a justification for only including 6 principal components?

Is there a reason that the chromosomes X and Y were excluded?

Line 269: ‘BOLT-RMEL’ should be ‘BOLT-REML’

Figures:

Figure 1b needs better labelling – it is currently very confusing to understand.

Reviewer #2: This study intended to find genetic contributors for ten inflammation markers by performing GWAS of two samples. They found and replicated 16 associated genomic regions, of which four are novel. Further, they estimated SNP-based heritability ranging from 0 for EPO up to 73% for S100B. Finally, the authors mapped these associated variants to enhancer elements, suggesting a possible transcriptional effect of genomic variants on the inflammatory markers. Overall, this is a well-designed and conducted study with many strengths and merits. However, the major concern of this paper is about its writing. In many parts, description is over simplified, which made this paper hard to follow. It would be important and essential for the authors to expand almost all parts of the paper.

Reviewer #3: The manuscript is well-written and provides important insights into genetics of inflammatory biomarkers. The results contain interesting genetic instruments for use in future Mendelian randomization studies. I have few questions related to manuscript.

1) The biomarkers were measured from dried blood spots. It is well known that for example heparin releases cytokines from receptors, such as ACKR1 (DARC), which serves as cytokine reservoir on red blood cell surface. This release induced by heparin might mask some genetic signals which would be detected if quantification would have been done by using non-heparin treated blood samples. Is there any previous data on how cytokine measures done from dried blood correlates with measures done from plasma or serum?

2) Could you explain why you calculated SNP heritability explained by each chromosome by excluding the pertinent chromosome from SNP heritability estimation and then subtracting the obtained SNP heritability from total SNP heritability explained by all autosomes instead of just calculating SNP heritability for one chromosome at a time?

3) According to Skogstrand et al. inter-assay variability for S100B was over 13%. Were cytokines assayed independently from genotyping batches?

4) S100B quantification was done with in-house developed platform. Do you know what this approach actually measures? Can capture antibody block the binding of detection antibody?

5) The Supplementary tables don't have any foot notes or explanation where the data originated from. Therefore, it is hard to track what data they actually contain. Could you describe these little more specifically on each spreadsheet?

6) According to the methods each genotyping wave was imputed separately. Sample size is an important determinant of phasing accuracy and this accuracy decreases rapidly when sample size drops below 1000 which in turn impairs imputation accuracy. What was the sample size in each genotyping wave?

**Have all data underlying the figures and results presented in the manuscript been provided?**

Reviewer #1: Yes

Reviewer #2: Yes

Reviewer #3: Yes

PLOS authors have the option to publish the peer review history of their article (what does this mean?). If published, this will include your full peer review and any attached files.

Reviewer #1: **Yes: **Dr Ruth E Mitchell

Reviewer #2: No

Reviewer #3: No

---

## [Decision Letter · Decision Letter 1]

29 Sep 2020

Dear Dr wang,

We are pleased to inform you that your manuscript entitled "Genome-Wide Association Study Identifies 16 Genomic Regions Associated with Circulating Inflammatory Markers at Birth" has been editorially accepted for publication in PLOS Genetics. Congratulations!

Yours sincerely,

Caroline Relton, PhD

Associate Editor

PLOS Genetics

Hua Tang

Section Editor: Natural Variation

PLOS Genetics

Comments from the reviewers (if applicable):

Reviewer's Responses to Questions

**Comments to the Authors:**

Reviewer #1: The authors have provided very clear explanations to my comments.

Reviewer #2: The authors have addressed my concerns on this paper.

Reviewer #3: The authors provided satisfactory responses for previous comments. I have no further comments. Thank you for making summary statistics available.

**Have all data underlying the figures and results presented in the manuscript been provided?**

Reviewer #1: Yes

Reviewer #2: Yes

Reviewer #3: Yes

PLOS authors have the option to publish the peer review history of their article (what does this mean?). If published, this will include your full peer review and any attached files.

Reviewer #1: **Yes: **Ruth E Mitchell

Reviewer #2: No

Reviewer #3: No

**Data Deposition**

http://datadryad.org/submit?journalID=pgenetics&manu=PGENETICS-D-20-00241R1

**Press Queries**

---

## [Editor Report · Acceptance letter]

30 Oct 2020

PGENETICS-D-20-00241R1 

Genome-Wide Association Study Identifies 16 Genomic Regions Associated with Circulating Cytokines at Birth 

Dear Dr wang, 

We are pleased to inform you that your manuscript entitled "Genome-Wide Association Study Identifies 16 Genomic Regions Associated with Circulating Cytokines at Birth" has been formally accepted for publication in PLOS Genetics! Your manuscript is now with our production department and you will be notified of the publication date in due course.

With kind regards,

Laura Mallard

PLOS Genetics

On behalf of:
